

# Fluorescent biological aerosol particles over the central Pacific Ocean: covariation with ocean-surface biological activity indicators

Kaori Kawana[1], Kazuhiko Matsumoto[1], Fumikazu Taketani[1], Takuma Miyakawa[1], Yugo Kanaya[1]

5   [1] Earth Surface System Research Center, Research Institute for Global Change, Japan Agency for Marine-Earth Science and Technology (JAMSTEC), Yokohama, 2360001, Japan

*Correspondence to*: Kaori Kawana (kawanak@jamstec.go.jp)

**Abstract.** Combining Waveband Integrated Bioaerosol Sensors and DNA staining techniques, online and offline shipboard observations of fluorescent aerosol particles in the atmosphere were carried out over the central Pacific Ocean during March 2019 to identify bioaerosols and determine their spatio-temporal distribution. To understand the origins of and processes associated with bioaerosols, we conducted correlation analyses of fluorescent particle number concentration, wind speed, and a variety of chemical and biological indicators, including concentrations of chlorophyll *a*, bacteria, marine organic gel particles such as Transparent Exopolymer Particles (TEPs) and Coomassie Stainable Particles (CSPs). Five-day backward trajectory analysis indicated that oceanic air masses were dominant between 6 and 18 March after which the influence of long-range transport from the continent of Asia was prominent. For the first period, we identified certain types of fluorescent particles as bioaerosols with marine origins, because their number concentrations were highly correlated with concentrations of TEPs and bacteria (R: 0.80–0.92) after considering the wind speed effect. For the second period, there was strong correlation between another type of fluorescent particles and CSPs irrespective of wind speed, implying that the fluorescent particles advected from land were mixed with those of marine origins. From the results of our correlation analysis, we developed equations to derive atmospheric bioaerosol number density in the marine atmosphere over the central Pacific Ocean from a combination of biogenic proxy quantities (chlorophyll *a*, TEPs and bacteria) and wind speed. We conclude that it is likely that TEPs were transported from the sea surface to the atmosphere together with bacteria to form fluorescent bioaerosols.

## 1 Introduction

25   Biological particles derived from marine and terrestrial organisms, including viruses, fungi, bacteria, pollen, and their fragments (Fröhlich-Nowoisky et al., 2016; Huffman et al., 2020) could represent a large proportion of the mass concentration of coarse particles in the atmosphere (Jaenicke, 2005). Originating from the marine ecosystem, organic matter in the surface seawater is uplifted by wind in the course of sea spray aerosol (SSA) formation; these biological particles could affect the cloud systems by acting as cloud condensation nuclei and ice nucleating particles (INPs) (Wilson et al., 2015; Šantl-Temkiv et al., 2020). Previous studies have shown that ice nucleation activity occurs at higher temperatures on biological materials



than on other ice nucleation active particles (e.g., mineral particles, dust, and volcanic ash) (Hoose et al., 2012; Murray et al., 2012). For example, Hoose et al. (2012) reported that freezing occurs at above −10 °C for bacteria, whereas other types of INPs begin to freeze only at around −30 °C. Mason et al. (2015) reported that biological particles were the major contributor of INPs in a coastal area with a temperature range of −15 to −25 °C, while the contribution of non-biological particles was

larger at −30 °C. Therefore, biological particles may be important as INPs in the formation of ice clouds at high latitudes and also in the formation of mixed phase clouds over the midlatitudes or even in cirrus cloud formation over the tropical regions. However, the importance of biological particles relative to dust as INPs has yet to be fully proven, partly because the origins, abundance, and roles of marine bioaerosols are poorly characterized; such information is needed to support model results (Burrows et al., 2009; 2013).

There are several methods to detect biological particles. Autofluorescence, which involves exciting and detecting fluorescent chromophores such as amino acids, proteins, and coenzymes is an effective method (e.g., Pöhlker et al., 2012). Recently, Wideband Integrated Bioaerosol Sensors (WIBS) and Ultraviolet Aerodynamic Particle Sizers (UV-APS) have been developed for online measurement and analysis of fluorescent particles (Pöschl et al., 2010; Gabey et al., 2010; Gosselin et al., 2016). Spectral features of single particle fluorescence have also been used to detect biological particles (Taketani et al., 2013;

Könemann et al., 2019). However, because of interference from other types of fluorescent particles (e.g., Polycyclic Aromatic Hydrocarbons (PAHs)), clear differentiation is necessary. Accurate, non-real-time methods such as nuclear staining with fluorescence microscopy and genetic phylogenetic search with Polymerase Chain Reaction (PCR) have also been used (Maki et al., 2013; Fröhlich-Nowoisky et al, 2016). Several methods to detect biological substances in seawater related to bioaerosols have also been developed. For example, staining and light absorption measurements have been used to detect gel-like organic

particles such as polysaccharide-containing Transparent Exopolymer Particles (TEPs) produced from phytoplankton exudations and protein-containing Coomassie Stainable Particles (CSPs) from the degradation of dead cells. These substances, as well as small cells or bacteria, are thought to be directly transported into the atmosphere via the formation of sea spray particles (Wurl et al., 2008; Engel et al., 2016; Thornton et al., 2018) or at least play key roles in the formation of bioaerosols. However, few field studies have simultaneously analyzed aerosol and seawater compositions to directly examine the link

between these components.

In this study, we report observations conducted during the MR18-06 Leg4 cruise of RV *Mirai* in March 2019 to examine the spatiotemporal distribution of biological aerosol particles over the central Pacific between Tahiti and Japan and the link between the bioaerosols and their potential oceanic precursors or proxies. Our focus is on the geographical distributions of fluorescent particles (both autofluorescent and epifluorescent), biogenic organic gel substances (TEPs and CSPs), other

relevant biological indicators (e.g., chlorophyll a (Chl-*a*) and bacteria) of the surface seawater, and their correlations. On the basis of our results, we discuss the link between marine substances or proxies and bioaerosol formation, and develop equations to provide best estimates of bioaerosol number density over the study region.



## 2 Cruise Observations

Observations were conducted between 6 and 25 March 2019 over the central Pacific Ocean from Papeete, Tahiti (17.32° S, 149.34° W) to Shimizu, Japan (35.02° N, 138.30° E) onboard the research vessel *Mirai*. The cruise track is shown in Fig. 1. For the fluorescent aerosol particles (FAPs), Wideband Integrated Bioaerosol Sensor Model 4A (WIBS-4A, Droplet

Measurement Technologies, Longmont, CO, USA) was used for continuous online measurements and Bioplorer (KB-VKH01, Koyo Sangyo Co., Ltd., Tokyo, Japan) (Nishimura et al., 2006) was used for offline, sampling-based onboard measurements. For WIBS-4A, aerosol particles were sampled from an inlet on the compass deck via conductive tubing and introduced into the instrument at a flow rate of 2.5 L min$^{-1}$ (sample: 0.3 L min$^{-1}$, sheath: 2.2 L min$^{-1}$). The WIBS-4A recorded autofluorescence of individual particles per excitation wavelength on two fluorescence detector channels (310–400 nm and 420–650 nm); xenon

flash lamps emitting at two excitation wavelengths (280 and 370 nm) were used (Healy et al., 2012). FAPs were classified into seven types according to the fluorescence patterns; those with fluorescence in a single wavelength range were defined as Type A (excitation=280 nm, fluorescent=310–400 nm), Type B (excitation=280 nm, fluorescent=420–650 nm), or Type C (excitation=370 nm, fluorescent=420–650 nm). The particles emitting fluorescence in multiple wavelength ranges were classified as Types AB, AC, BC, or ABC (Perring et al., 2013). Size distributions of fluorescent and non-fluorescent particles

were derived from the scattering intensity measurements obtained with a continuous-wave 635 nm diode laser (O'Connor et al., 2013). Previous studies have proposed determining 1σ—the background fluctuation in the absence of particles—and using 3σ or 9σ signal levels as a baseline threshold to distinguish between fluorescent and non-fluorescent particles (Crawford et al., 2015; Perring et al., 2018). In this study, we used 3σ as the threshold value. Fluorescent Polystyrene latex particles of 2 μm (PSL, G0200, Thermo Fisher Scientific, Waltham, MA, USA) were introduced before and after the observations to check the

validity of the particle size and fluorescent intensity for the instrument (Robinson et al., 2017; Savage et al., 2017). The samples for Bioplorer were collected with a flow rate of ~1 L min$^{-1}$ directly onto gold-coated membrane filters (pore size: 0.4 μm, KB-VKF02, Koyo Sangyo Co., Ltd., Tokyo, Japan) where epifluorescence detection was conducted. Bioplorer was used as a fluorescence microscope to detect biological particles with higher selectivity, and the counting of fluorescent spots upon UV excitation was automated. Apparatus performance has been previously validated using marine bacteria (Nishimura et al., 2006;

2008). The obtained samples were stained with 4',6-diamidino-2-phenylindole (DAPI, KB-VKR01 and KB-VKR03, Koyo Sangyo Co., Ltd., Tokyo, Japan) and Hoechst 33342 (Dojindo Laboratories, Mashiki, Japan) to increase the efficiency in biogenic fluorescent particle detection. After staining for two minutes, the samples were rinsed with Milli-Q water three times, and then measured with Bioplorer.

For chemical composition analysis, a high-volume air sampler (HVS, 120SL, Kimoto Electric Co., Ltd., Osaka, Japan) was

30 installed on the deck and particle less than 2.5 micrometers in diameter (PM$_{2.5}$) were collected onto a quartz filter using an impactor (HVI-2.5, Tokyo Dylec Co., Tokyo, Japan) at a flow rate of ~740 L min$^{-1}$ during 2–3 days. To prevent contamination from ship exhaust, the pump stopped automatically when the wind direction deviated more than ±75° from the bow direction or when the wind speed fell below 2 m s$^{-1}$. The samples were stored at −20 °C in a freezer. The mass concentrations of ionic



species ($NH_4^+$, $Na^+$, $K^+$, $Ca^{2+}$, $Mg^{2+}$, $Cl^-$, $NO_3^-$, and $SO_4^{2-}$) in $PM_{2.5}$ were obtained by ion chromatography (ICS-1000, Dionex Co., CA, USA). The mass concentrations of organic carbon (OC) and elemental carbon (EC) in $PM_{2.5}$ were obtained using a thermal/optical carbon analyzer (DRI model 2001, Desert Research Institute, Reno, NV, USA) with the Interagency Monitoring of Protected Visual Environments (IMPROVE) protocol. The size distribution of aerosol particles was measured

with an optical particle counter (OPC, KR-12A, Rion, Kokubunji, Japan). Ozone ($O_3$) and carbon monoxide (CO) concentrations were also measured with UV (Model 49C, Thermo Fisher Scientific, Waltham, MA, USA) and nondispersive infrared sensors (Model 48C, Thermo Fisher Scientific, Waltham, MA, USA) (Kanaya et al., 2019). To avoid contamination from ship exhaust, the data points from the online measurements were screened using the same criteria that were applied to the operation of the pump of the high-volume air sampler.

Surface seawater sampling for investigation of TEPs, CSPs, phytoplankton pigments, and nutrients concentration was conducted using a bucket at 15 stations in the cruise (Table S1). For the analysis of TEPs and CSPs, seawater samples of 200 mL were filtered onto Whatman 0.4 μm Nuclepore hydrophilic polycarbonate membrane filters (Cytiva, Tokyo, Japan) where particles collected. For TEPs, 1 mL of Alcian blue staining solution, adjusted to pH 2.5, was added to the filter and the filter was rinsed three times with 1 mL of Milli-Q water after 4 seconds of staining. Filters were soaked for 2–5 h in 6 mL of 80 %

sulfuric acid to elute the dye and the absorbance of the solution was measured at a wavelength of 787 nm. The calibration curve was produced using a xanthan gum solution (XG, Sigma-Aldrich Co. LLC, St. Louis, MO, USA) as a standard before and after observation, and the TEP concentrations were reported as XG equivalent (Passow and Alldredge, 1995). For CSPs, 1 mL Coomassie Brilliant Blue staining solution was added to the filter, which was then rinsed five times with 1 mL of Milli-Q water after 1 minute. Filter samples were soaked for 2 h in 4 mL of 3 % sodium dodecyl sulfate in 50 % isopropyl alcohol

with ultrasonic extraction to elute the dye and the absorbance of the solution was measured at a wavelength of 615 nm. The calibration curve was calculated using bovine serum albumin (BSA, Sigma-Aldrich Co. LLC, St. Louis, MO, USA) as a standard before and after observation, and the CSP concentrations were reported as BSA equivalent (Cisternas-Novoa et al., 2014). Seawater samples were filtered onto a Whatman GF/F filter (Cytiva, Tokyo, Japan) and extracted in N, N-dimethylformamide for the measurement of phytoplankton pigments. The measurements were conducted with a fluorometer

(model 10-AU, Turner Designs, Inc., San Jose, USA) for Chl-*a*, and a high-performance liquid chromatography (HPLC) system (Agilent, Santa Clara, CA, USA) for biomarker pigments. The relative contribution of each phytoplankton group was calculated by using the chemotaxonomy program CHEMTAX (Mackey et al. 1996) with the initial pigment ratios shown by Araujo et al. (2017) compiled for the subtropical region. Nutrient analyses were performed using a continuous segmented flow analyzer (QuAAtro 2-HR, BL TEC K.K., Tokyo, Japan). Meteorological parameters at the sea surface, such as wind speed

(WS) and sea surface temperature (SST), measured by R/V *Mirai* monitoring system were employed for the analysis.

Surface seawater samples were also collected at the same time and stored in a freezer at −20 °C for bacteria number density measurements. Abundances of marine bacteria including both autotrophic picocyanobacteria and heterotrophic bacteria were determined. The samples were collected using a bucket, fixed with glutaraldehyde to a final concentration of 1 % and preserved at −80 °C until analysis. Marine bacteria were stained with SYBR Green I DNA stain (Thermo Fisher Scientific, Waltham,



MA, USA) for 15 minutes in the dark (Marie et al., 1997), and counted using flow cytometry (EC800, Sony Biotechnology Inc., Japan) against the side scatter signal from the green fluorescence.

## 3 Results and Discussion

### 3.1 Trajectories, gases, and aerosol chemical composition

Five-day backward trajectories of air parcels were calculated using NOAA's HYSPLIT model (Stein et al., 2015) from a starting altitude of 500 m to classify the observed air masses. The results along the cruise track (Figs. 1a and 1b) indicate that oceanic air masses were dominant from 6 to 18 March 2019 (Period 1) and long-range transport from the continent of Asia was also prominent between 19 and 25 March 2019 (Period 2). Figure 1c showed the location of the ship during the cruise observation. Figure 2 shows the time series of 1h averaged meteorological parameters (temperature, relative humidity, wind

direction, and wind speed) and $O_3$ and CO concentrations. The $O_3$ concentration increased gradually (to ~30 ppb) after 13 March when the ship entered the Northern Hemisphere and increased again to ~45 ppb on 19 March, while the CO concentration increased slightly on 19 March and then increased considerably at the end of the observation period (22–23 March). These results support our classification of air masses into Periods 1 and 2.

    Figure 3 shows the mass concentrations and mass fractions of EC, OC, inorganic components, and sea salt (SS) from the

$PM_{2.5}$ samples. Mass concentrations of SS and non-sea salt sulfate (nss-sulfate) were calculated using standard seawater composition equations (Warneck, 1999) and $Na^+$ concentrations. OC, SS, and nss-sulfate were the major components of our samples. Mass fractions of OC, SS, and nss-sulfate were 38 %, 30 %, and 25 %, respectively during Period 1 when oceanic air masses were dominant, and were 48 %, 18 %, and 26 %, respectively, during Period 2. The high concentrations of OC and SS in the oceanic air mass suggest that organic matter from marine ecosystems at the sea surface may have been ejected into the

atmosphere with sea spray particles under the high wind conditions on 12–14 March 2019.

### 3.2 Temporal variation and types of fluorescent aerosol particles

    Figure 4 shows the temporal variation of the 1h averaged number concentrations of FAPs and total (fluorescent and non-fluorescent) particles larger than 1 μm measured with WIBS-4A (Fig. 4a) and the relative fractions of classified fluorescent particles (Fig. 4b). The average number concentrations of the FAPs (and their fractions relative to the total number

concentrations of particles) were 35 particles $L^{-1}$ (1.8 %) for the entire study period, 30 particles $L^{-1}$ (1.3 %) for Period 1, and 50 particles $L^{-1}$ (2.6 %) for Period 2. Eighty nine percent of the FAPs belonged to Types A (62 %), B (20 %), or C (7 %), which emitted fluorescence in a single wavelength band. For Period 2, the fractions of Type B particles (30 %) and the types of particles emitting fluorescence in multiple bands (AB, AC, BC, and ABC; 19 %) were higher than those for Period 1 (B: 17 %, AB+AC+BC+ABC: 9 %), implying that the fluorescent properties of particles in Periods 1 and 2 were essentially

different (Fig. 4b). Figure 4c shows strong correlation between the time series of Types A and C particles (R: 0.74), and weak correlations between Types B and A, and between Types B and C (R≈0.3). A previous study characterizing fluorescence





patterns of a variety of bioaerosols (bacteria, fungi, spore, and pollen) using laboratory experiments showed that Type A particles originated from bacterial and fungi species (Hernandez et al., 2016). Similar results were found with marine bacteria (Mitts et al., 2019). Therefore, we infer that bacteria (and fungi, if present) at the sea surface were transported into the atmosphere by wind and were detected as fluorescent particles in the atmosphere.

Number size distributions of fluorescent particles have been reported for different marine and terrestrial environments. At forest, mountain, and urban terrestrial sites, previous studies have commonly suggested the dominance of super-micron (2–5 µm) fluorescent particles (Gabey et al., 2010; 2011). Results obtained above the ocean are more diverse; Wilson et al. (2015) suggested the dominance of relatively small particles around 1 µm while Creamean et al. (2019) indicated that coarse particles of 2–4 µm were dominant; Mason et al. (2015) reported that both fine (Dp<1.0 µm) and coarse (1.8–3.2 µm) mode particles

were present. Our results show that most FAPs had a peak at 1–2 µm (Fig. S1) with the exception of Type ABC particles, indicating that the FAPs may have mainly consisted of relatively small particles that have yet to experience aggregation or growth. These small FAPs may be related to the most of marine bacteria detected by flow cytometry, which also exhibited as <2 µm. Correlations of the time series of fluorescent particle abundance with biogenic indicators in surface seawater are discussed in Sect. 3.4 in detail.

The emission of primary particles and the mass fraction of organic matter from the sea surface increase with wind speed (Carlson, 1983; Gantt et al., 2011). Our results also show correlation between the number concentration of FAPs and wind speed. On 12 March, the number concentration of FAPs increased from ~10 to ~30 particles $L^{-1}$ when wind speed increased from ~4.8 to 13.5 m $s^{-1}$ (Fig. 4c). The number concentrations of ambient particles with size ranges of 0.3–0.5, 0.5–0.7, 0.7–1, 1–2, 2–5, and 5.0 µm measured with OPC (Fig. 4d) also increased after 12 March 2019. The mass fraction of SS was

particularly high (56 %), and SS and OC accounted for 90 % of the mass for this period. The ratio of large particles (Dp>2.0 µm) to small particles (Dp<2.0 µm) increased considerably (Fig. 4e). These results indicate that organic matter, and FAPs in particular, were transported efficiently from the ocean surface to the atmosphere under high wind conditions.

    Figure 5 shows the number concentrations of Types A, B and C particles (Fig. 5a), and Types AB, AC, BC, and ABC particles (Fig. 5b) from WIBS-4A and Bioplorer (DNA nuclear staining method) measurements. The correlation coefficients between

the number concentrations obtained from the two methods were very high (R>0.79, Table 1) for Types A, B and C particles during both Periods 1 and 2 and under high and low wind speeds. The strong correlation between the number concentrations of fluorescent particles from WIBS-4A and biogenic fluorescent particles identified by DNA nuclear staining confirms online measurements based on autofluorescence as a reliable method of bioaerosol detection. Figure 5a shows that the total number concentrations of Types A, B, and C particles from WIBS-4A were almost in the same range as those of biological particles

from Bioplorer. Figure 5b shows that number concentrations from WIBS-4A were smaller than those from Bioplorer for the particles emitting fluorescence in multiple bands (Types AB, AC, BC and ABC). Given the large uncertainties associated with the differences between the two measurement methods, such as detected size range and detectivity near the 3σ threshold, it is difficult to have a meaningful discussion about differences between measured values when the values are within a factor of ~2 of each other.



### 3.3 Concentrations of nutrients, Chl-*a*, bacteria, and organic substances in the surface seawater

To characterize the oceanic conditions in the study region, time series of concentrations of nutrients (nitrate, ammonium, and phosphate), Chl-*a*, bacteria, TEPs and CSPs from the surface seawater are shown in Figure 6. We identified four different
regions or periods where there were large contrasts between nutrient and Chl-*a* concentrations: The South Pacific subtropical region (SP), equatorial upwelling region (EQ), North Pacific subtropical region (NP), and south of the Kuroshio Extension (KR). Nutrient concentrations were high in the equatorial upwelling region, low in the North Pacific subtropical region, and high again in the south of the Kuroshio Extension on 22 March. Concentrations of Chl-*a* and CSPs also responded to two of the peaks in nutrient concentration (Fig. 6a), with large increases in the south of the Kuroshio Extension and smaller increases
in the equatorial upwelling region, but with different magnitudes (Figs. 6b, 6e). On the other hand, variations of bacteria and TEP concentrations were different from those of Chl-*a* and CSP concentrations; with high concentrations in the equatorial upwelling region and no large increases in the south of the Kuroshio Extension (Figs. 6c, 6d). In general, the biological activity of marine ecosystems with primary production is highly dependent on nutrients and SST (Engel et al., 2015). However, our results indicate that factors controlling Chl-*a* and CSP variations were different from those controlling bacteria and TEP
variations. The correlations between concentrations of marine biota (represented by Chl-*a* and bacteria) and biogenic organic particles (represented by TEPs and CSPs) are shown in Fig. S2. Chl-*a* is a direct indicator of phytoplankton biomass and is therefore widely used to assess biological activities of marine ecosystems in studies involving in situ and satellite observations. While previous studies reported a good correlation between Chl-*a* (indicating the presence of phytoplankton) and TEPs, which are produced from phytoplankton exudations (Wurl et al., 2008; Zamanillo et al., 2019), our results indicate only moderate
correlation (R≈0.4) because TEP production may be enhanced under nutrient-limited condition as opposed to primary production. It has been reported that in case of some diatoms and also picocyanobacteria, TEP production increases in the nutrient-poor waters (Passow, 2002; Gärdes et al., 2012, Deng et al. 2016). The values of ratio of TEP/Chl-*a* in this study (Figure 6d) was actually higher in the North Pacific subtropical region, suggesting that phytoplankton in the nutrient-limited condition had higher TEP production per cell. By contrast, CSP concentration was strongly correlated with Chl-*a* concentration
(R: 0.81), particularly in the south of the Kuroshio Extension, while the correlation between CSP and TEP concentrations was weak (R: 0.43). This suggests that CSPs may be governed by factors and cycling dynamics that are different from those for TEPs (e.g., degradation).

### 3.4 Association of marine biota to the formation of bioaerosols in the atmosphere

We identified the parameters indicating biological activity (Chl-*a* and bacteria) or related substances (TEPs and CSPs) in the surface seawater that are associated with the abundance of FAPs as a proxy of bioaerosols. The association would indicate either the substances from surface seawater directly participated in bioaerosol emission processes and thereby integrated into





the generated bioaerosols or that the substances are just useful as proxies to describe bioaerosol abundance. Because such proxies can be of use in the parameterization of numerical models, we developed equations to derive bioaerosol abundance from several oceanic parameters. The number of primary particles, especially organic matter, released from the ocean surface, is generally known to be influenced by the ocean surface environment, wind speed (WS), and SST (Gantt et al., 2012). The

5 SSA flux is often approximated with a power law equation with WS at the altitude of 10 m ($U_{10}$) and an exponent of 3.41 (Monahan et al., 1986; de Leeuw et al., 2011) or 3.5 (Grythe et al., 2014). There have been fewer studies of power law equations of SSA number concentrations in the atmosphere at the sea surface. These studies have reported exponent ranges of 0.68 (observations) to 1.62 (model) (Jaegle et al., 2011), 2.1 to 2.8 (Ovadnevaite et al., 2012), and 2.8 (Saliba et al., 2019). To consider the effect of wind speed on particle number concentration, we assumed an exponent of 1 in this study. When an

10 exponent of 2 was used, the results remained almost unchanged (see Fig. S4 and bottom two lines of Table 2).

Our analysis included the major types of bioaerosols measured with WIBS-4A. Because our results show a strong correlation between Types A and C particles, we analyzed all Types A and C particles together (Type A+C). We also analyzed Type B particles. Figures 7a–7j show the temporal variations of the number concentrations of the bioaerosols (Types A+C and B) measured in the WIBS-4A, wind speed, and concentrations of different bioindicators: Chl-a, bacteria, TEPs, and CSPs. There

was a robust positive correlation between number concentrations of Type A+C particles and WS; the correlation coefficient even increased to 0.85 for Period 1 when the oceanic air mass was dominant (11 data points are available for this period) (Figs. 7a and 8a and Table 2). By contrast, correlation between number concentration of Type B particles and WS was weak (R: 0.36). The number concentrations of bioaerosols showed a weak negative correlation with SST; Saliba et al. (2019) reported a clear dependence of total number concentration of SSAs on temperature, which is absent in our results (Fig. 8b).

Figures 8c–8f show correlations between number concentrations of atmospheric bioaerosols measured with WIBS-4A and products of wind speed and concentrations of different bioindicators. Number concentrations of Type A+C particles showed strong correlations with the product of WS and bacteria concentration (R: 0.80) and the product of WS and TEP concentration (R: 0.85); these trends are similar to those found in the correlations between particle number concentrations and WS only. Correlation coefficients between number concentrations of Type B particles and the product of WS and bacteria concentration

(R: 0.83) and the product of WS and TEP concentration (R: 0.92) were higher than correlation coefficients between number concentrations of Type B particles and WS only (R: 0.36), or bacteria concentration only (R: −0.26), or TEP concentration only (R: −0.62) (Fig. S3, Table 2). These results suggest that TEPs and bacteria were the major components associated with the formation of atmospheric bioaerosols over the ocean. Wind uplifts the organic matter present in the ocean surface layer into the atmosphere to form bioaerosols composed of biogenic organic matter (e.g., TEPs) or their aggregates with bacteria

(Aller et al., 2017). On the basis of the data, they had collected during a cruise in the Pacific Ocean, and specifically in the south of the Kuroshio Extension near Japan, Hu et al. (2017) reported a high percentage (>89 %) of viable bacteria, which suggests the possibility of in situ formation of bioaerosols driven by wind acting on the sea surface. They also reported number concentrations of bacteria of 10–250 cells $L^{-1}$ in the atmosphere, which is roughly consistent with our measurements of Type A particle abundance in a similar region (5.0–31.0 particles $L^{-1}$, Fig. 5a). Flow cytometry studies reported bacteria



concentration levels in seawater of $(6–9)×10^5$ cells mL$^{-1}$ in the equatorial region, $(3–6)×10^5$ cells mL$^{-1}$ in the subtropical Pacific region (Landry and Kirchman, 2002), and $(2.3–7.4)×10^5$ cells mL$^{-1}$ in the subtropical Pacific region (Campbell et al., 1997). These results are also consistent with the bacterial abundance of $(2.5–5.1)×10^5$ cells mL$^{-1}$ that we found in our study.

We also found that the correlation coefficients between the number concentrations of Type B particles and the product of WS and Chl-*a* concentration (R: 0.32) or the product of WS and CSP concentration (R: 0.58) were larger than the correlation coefficients between the number concentrations of Type B particles and Chl-*a* or CSP concentration only (Table 2). Results from high-performance liquid chromatography indicate high abundance of picocyanobacteria such as Prochlorococcus and Synechococcus (Dp<2 μm) in the equatorial region (33–42 %) and the North Pacific subtropical region (63–89 %). By contrast, there was high abundance of other nano- and microplankton (2–20 μm and >20 μm, respectively) in the south of the Kuroshio Extension (36–47 %) during Period 2 (Fig. S5). Smaller phytoplankton containing Chl-*a* might have been directly uplifted by the wind and detected as bioaerosols in the subtropical and equatorial regions.

Correlation coefficients between number concentrations of Type A+C particles and product of WS and TEP concentration or product of WS and bacteria concentration were lower for the entire study period (R: 0.54–0.83) than for Period 1 alone (R: 0.80–0.85). A similar trend was found for the correlation coefficients between the number concentrations of Type B particles and product of WS and TEP concentration or product of WS and bacteria concentration (R: 0.26–0.41 for entire study period, R: 0.83–0.92 for Period 1). This suggests that components other than TEPs and marine bacteria may have contributed to the measurements obtained from WIBS-4A during Period 2. On the contrary, correlation coefficients between number concentrations of atmospheric bioaerosols (Types A+C or B) and product of WS and CSP concentration were higher for the entire study period (R: 0.60–0.83) than for Period 1 alone (R: 0.49–0.58). Number concentrations of Type B particles increased during Period 2 (Figs. 7 and 8), suggesting that in addition to TEP and bacteria, CSP also made large contributions as bioaerosols in the atmosphere. Unlike Period 1, there was little correlation between the temporal variation of fluorescent particle number concentrations near land and the increase or decrease of wind speed (Figs. 7a and 7f). Mayol et al. (2017) found that 33–68 % of fluorescent particles were of oceanic origin while the remainder was of terrestrial origin; these results apply even to areas over the Pacific Ocean that are far from continents and islands, suggesting possible long-range transport of bioaerosols of terrestrial origin in addition to in situ oceanic bioaerosol formation. The influence on the marine atmosphere of bioaerosols originating from continents and islands will be the subject of future studies.

Finally, from the results of our correlation analysis, we developed equations to derive the number concentrations of total atmospheric bioaerosols y (including Types A, B, C, AB, AC, BC, and ABC particles) in the atmosphere at the sea surface for Period 1 (Fig. 9):

y (particles L$^{-1}$) = 0.075 ·[TEP, μg XGeq L$^{-1}$] · WS (m s$^{-1}$) + 5.7 (R: 0.88)     (1)

y (particles L$^{-1}$) = 0.0052 ·[bacteria, cells μL$^{-1}$] · WS (m s$^{-1}$) + 9.4 (R: 0.80)  (2)

and

y (particles L$^{-1}$) = 3.68 · [Chl-*a*, mg m$^{-3}$] · WS (m s$^{-1}$) + 21 (R: 0.47) (3)

where [TEP], [bacteria], [Chl-*a*] represent the concentration of TEP, bacteria, and Chl-*a* in the surface seawater for Period 1.



Although the correlation coefficient between the number concentrations of atmospheric bioaerosols and Chl-*a* concentration was lower than that between the number concentrations of atmospheric bioaerosols and TEP or bacteria concentration, Eq. (3) would be more useful than Eqs. (1) and (2), as Chl-*a* data are much more easily available than data on TEPs or bacteria. Considering the difference between the number densities derived from WIBS-4A and Bioplorer (see Sect. 3.2), we tentatively suggest an uncertainty range of a factor of ~2 for Eqs. (1)–(3). These equations can be used to derive bioaerosol number concentrations over the remote Pacific Ocean to study the composition of organic aerosols and also to validate atmospheric chemistry models where other parameterizations are used. With further clarification of the relationship between INP and bioaerosol number densities, INP number densities may also be derived from these equations.

The relationship between atmospheric bioaerosols over the ocean and their biogenic sources for the different oceanic regions and meteorological conditions in this study are summarized as follows:

(1) Equatorial upwelling region (6.35° S–9.25° N, 9–13 March, 2019): High nutrient and Chl-*a* concentrations led to an increase in the formation of TEPs and CSPs. However, there was no correlation between the concentrations of biogenic materials and the number of bioaerosols in the atmosphere, suggesting that organic matter remained in the surface seawater, perhaps in the sea surface microlayer, instead of being transported into the atmosphere because of low wind speeds (between 6 and 11 March).

(2) North Pacific subtropical region (13.00° N–27.00° N, 14–20 March, 2019): Low nutrient concentrations resulted in low concentrations of biogenic substances and organic matter, but the abundance of TEP production per cell was high and the fraction of smaller picocyanobacteria was also high. As a result, high wind speeds resulted in an increase in the number concentrations of bioaerosols as wind-driven transportation of TEP and bacteria from the sea surface to the atmosphere occurred efficiently.

(3) South of the Kuroshio Extension (30.56° N–32.25° N, 21–22 March, 2019): The uplift of biogenic and organic substances from the ocean surface to the atmosphere likely occurred under the conditions of high nutrient concentrations and high wind speeds. The contribution of terrestrial fluorescent particles might have increased.

In this study, we have successfully linked the number concentrations of bioaerosols in the atmosphere to biogenic substances or biological activity indicators in the surface seawater by taking into account meteorological parameters over the open central Pacific. Fluorescence patterns of atmospheric particles and particle response to wind speed in oceanic air masses were different from those under the influence of terrestrial air mass. Different marine substances or biological activity indicators were found to be associated with the formation of bioaerosols. Our results suggest that TEPs aggregated with bacteria in the surface seawater could be transported into the atmosphere by wind to form bioaerosols, and that the bacteria can be detected as bioaerosols with fluorescence. Future comparative studies on the origin and behavior of bioaerosols should be conducted at sites under considerable terrestrial influence or where substances of mixed terrestrial and marine origins are present and well characterized.



## 4 Conclusions

During a research cruise over the central Pacific Ocean from Tahiti to Japan, we examined the spatio-temporal distribution of atmospheric fluorescent particles, and characterized the fluorescence patterns and number concentrations of bioaerosols. During the cruise, oceanic air masses were dominant between 6 and 18 Mar 2019 and the influence of terrestrial air masses was prominent between 19 and 25 Mar 2019. To identify potential precursors or proxies that are important to bioaerosol formation, we examined variations of Chl-*a*, bacteria, and biogenic gel organic particles (TEPs, CSPs) in the surface seawater. Number concentrations of autofluorescent particles as measured by WIBS-4A strongly correlated with wind speed, suggesting bioaerosols of oceanic origin. The dominant particles types were: Type A particles emitting fluorescence in the 310-400 nm range upon excitation at 280 nm, Type C particles emitting in the 420-650 nm range upon excitation at 370 nm (Types A+C, 69%), and Type B particles emitting in the 420–650 nm range upon excitation at 280 nm (20 %). The number concentrations of autofluorescent particles obtained from WIBS-4A agreed and covaried with those obtained from Bioplorer, which is an automated epifluorescence measurement system based on DNA staining, confirming the performance of WIBS-4A in detecting fluorescent biological particles originated from the marine biosphere.

The number concentrations of Types A+C and B particles were strongly correlated with the product of wind speed and TEP concentration in the surface seawater and the product of wind speed and bacteria concentration in the surface seawater. Concentrations of Chl-*a* and CSPs were also moderately correlated with number concentrations of both Types A+C and B particles in the oceanic air masses. When the influence of terrestrial air mass was prominent, the correlation between Types A+C and concentrations of TEPs or bacteria became weaker. On the contrary, the correlation between fluorescent particles (especially Type B particles) and CSP concentration was stronger over the entire study period than for only the period of oceanic air mass dominance. These results suggest that in addition to TEP and bacteria, CSP also made large contributions as bioaerosols in the atmosphere. We also developed equations of number concentrations of bioaerosols in the near-surface atmosphere as functions of wind speed and biological parameters (concentrations of Chl-*a*, bacteria, and TEPs) over the central Pacific Ocean.

**Data availability**

The observational data is available upon request to the corresponding author.

**Author contributions**

KK and YK designed the research. KK, KM, and FT performed the cruise observation, data collection, and data analysis with the contribution with TM and YK. KK wrote the manuscript and all co-authors provided comments to improve the manuscript.





**Competing interests**

The authors declare that they have no conflict of interest.

**Acknowledgments**

We acknowledge assistance from captain and crews of cruises and support from Marine works Japan, Ltd. and Nippon Marine

Enterprise, Ltd. for R/V *Mirai*. This research has been supported by the Ministry of Education, Culture, Sports, Science and

Technology (MEXT) and the MEXT/JSPS KAKENHI (grant number No.JP18H04143). We thank Tina Tin, PhD, from Edanz

Group (https://en-author-services.edanz.com/ac) for editing a draft of this manuscript.

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




**Table 1.** Correlation coefficients between number concentrations of fluorescent aerosol particles (FAPs) obtained by WIBS-4A (autofluorescent) and biological particles obtained by Bioplorer (DNA nuclear stain).

| Type | All Period | Marine |
|------|-----------|--------|
| A    | 0.84      | 0.86   |
| B    | 0.82      | 0.82   |
| C    | 0.79      | 0.83   |
| AB   | 0.86      | 0.86   |
| AC   | 0.09      | 0.09   |
| BC   | 0.61      | 0.61   |
| ABC  | 0.71      | 0.72   |
| All  | 0.89      | 0.89   |





**Table 2.** Correlation coefficients between number concentrations of fluorescent aerosol particles (FAPs) in the atmosphere and wind speed (WS) and/or bioindicator (I) concentrations when oceanic air masses were dominant.

| FAPs vs | Type | Wind | Chl-*a* | Bacteria | TEP | CSP |
|---------|------|------|---------|----------|------|------|
| I | A+C | 0.85 | −0.24 | −0.17 | −0.56 | −0.51 |
| | B | 0.36 | −0.45 | −0.26 | −0.62 | −0.52 |
| I * WS | A+C | | 0.45 | 0.80 | 0.85 | 0.49 |
| | B | | 0.32 | 0.83 | 0.92 | 0.58 |
| I * WS$^2$ | A+C | | 0.55 | 0.80 | 0.85 | 0.49 |
| | B | | 0.45 | 0.92 | 0.83 | 0.59 |





**Table 3.** Same as Table 2, but when the influence of terrestrial air masses was prominent.

| FAPs vs | Type | Wind | Chl-*a* | Bacteria | TEP | CSP |
|---------|------|------|---------|----------|------|------|
| I | A+C | 0.80 | 0.30 | −0.21 | −0.23 | 0.07 |
| | B | 0.76 | 0.73 | −0.25 | 0.00 | 0.63 |
| I * WS | A+C | | | 0.54 | 0.83 | 0.60 |
| | B | | | 0.26 | 0.41 | 0.83 |
| I * WS$^2$ | A+C | | | 0.71 | 0.79 | 0.63 |
| | B | | | 0.21 | 0.30 | 0.72 |




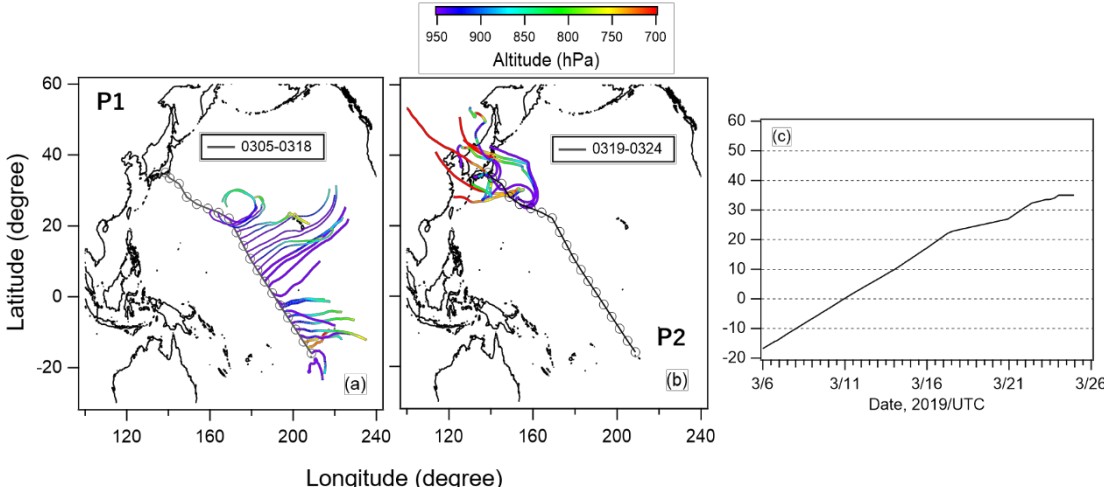

**Figure 1.** Five-day backward trajectories of air particles along with the cruise track initiated at 0600 and 1800 UTC of each day for (a) Period 1 (5–18 March 2019), and (b) Period 2 (19–24 March 2019). Open cycle markers represent the ship position during the observation. Color on trajectories show air parcel altitude. (c) Ship position during study period.





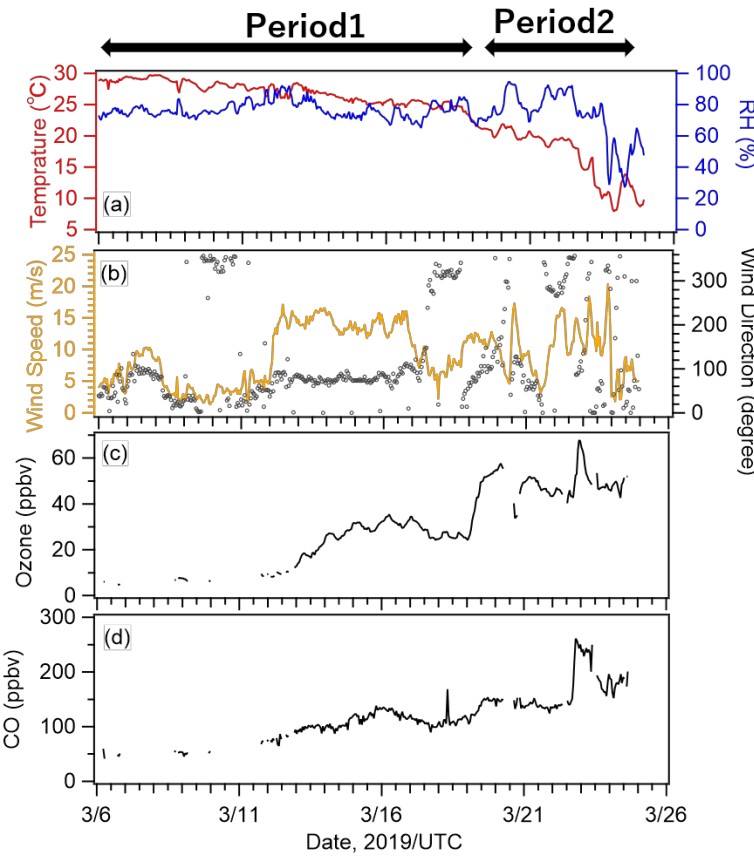

**Figure 2.** Time series of meteorological parameters: (a) air temperature and relative humidity, (b) wind speed and direction.

5    Time series of concentrations of (c) ozone and (d) carbon monoxide.




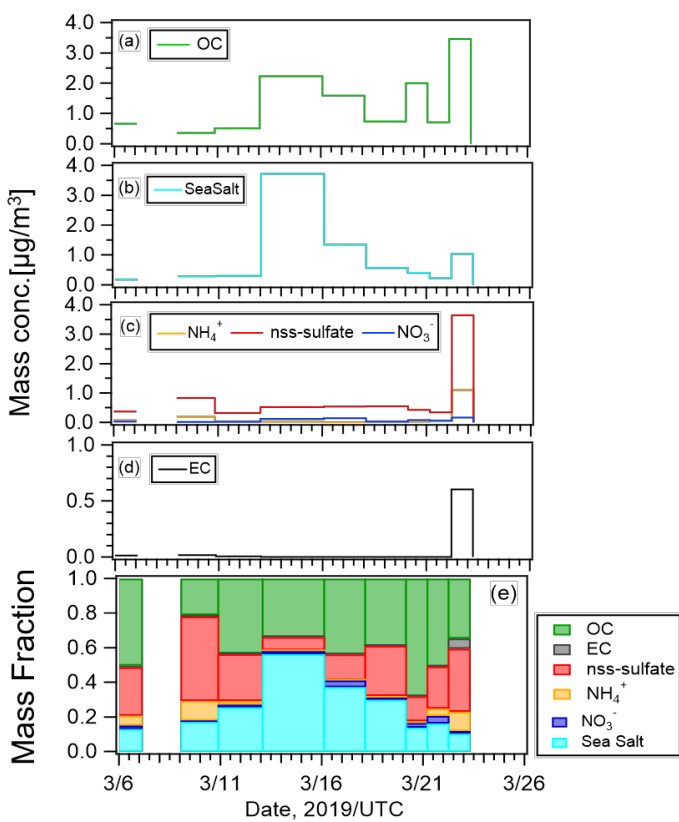

**Figure 3.** Mass concentrations of (a) organic carbon (OC), (b) sea salt, (c) inorganic compounds, (d) elemental carbon (EC). And (e) mass fractions of all components.

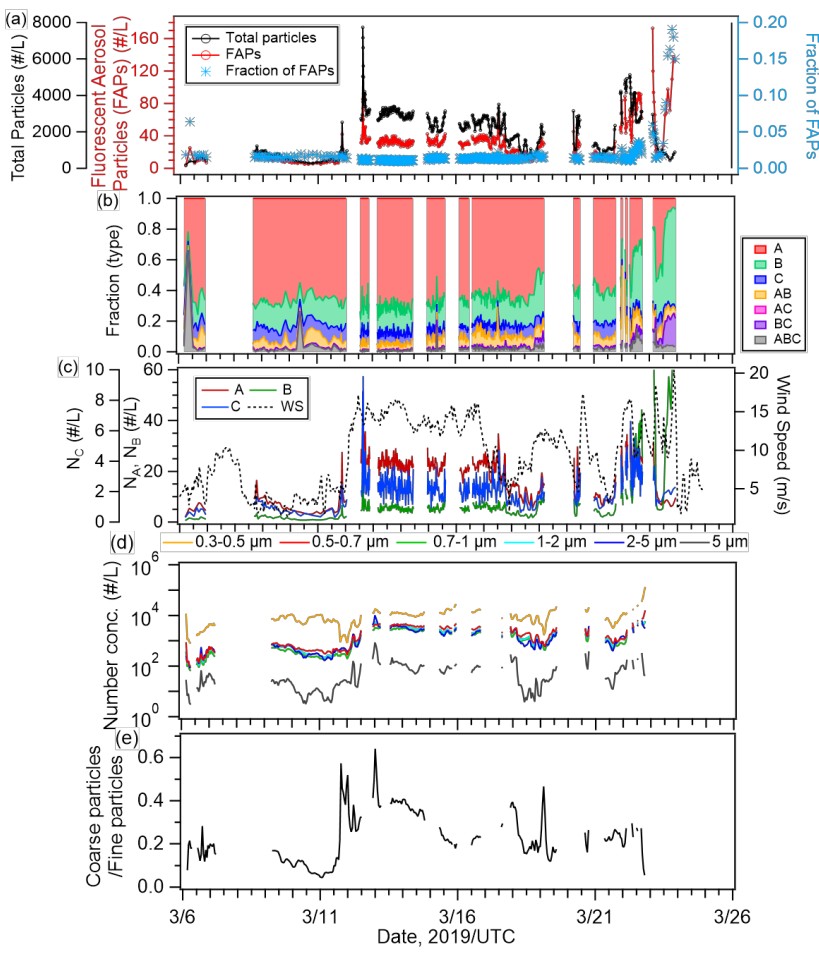

**Figure 4.** Time series of (a) number concentrations of all particles (black line) and fluorescent aerosol particles (FAPs, red line), and number fractions of FAPs (blue marker), (b) relative fractions of types, and (c) number concentrations of Types A, B, C particles and wind speed (dashed line). Time series of (d) size-resolved number concentrations of ambient particles, (e) ratio of number concentration of particles larger than 2 μm to number concentration of particles smaller than 2 μm.

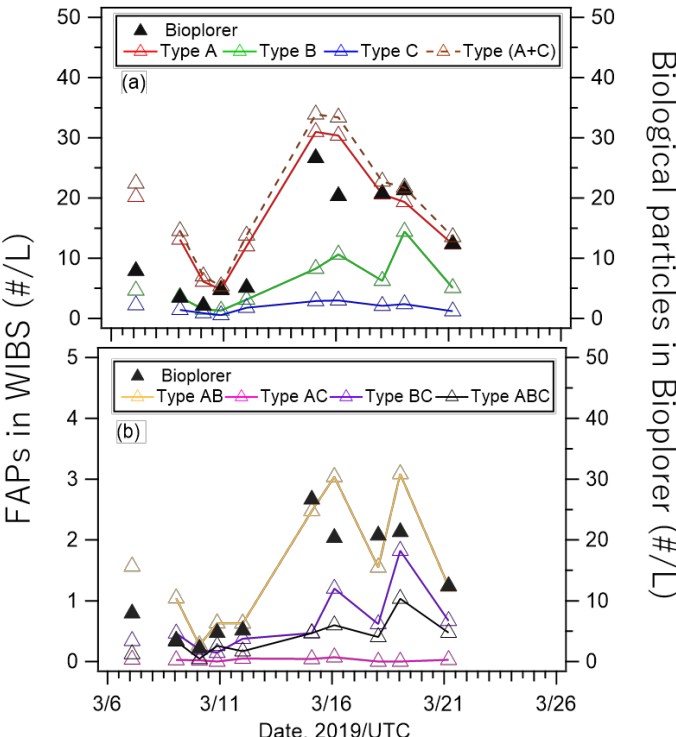

**Figure 5.** (a) Number concentrations of Types A, B, C and A+C fluorescent aerosol particles (FAPs), and (b) number concentrations of Types AB, AC, BC, and ABC fluorescent aerosol particles from WIBS-4A and biological particles from Bioplorer measurements.



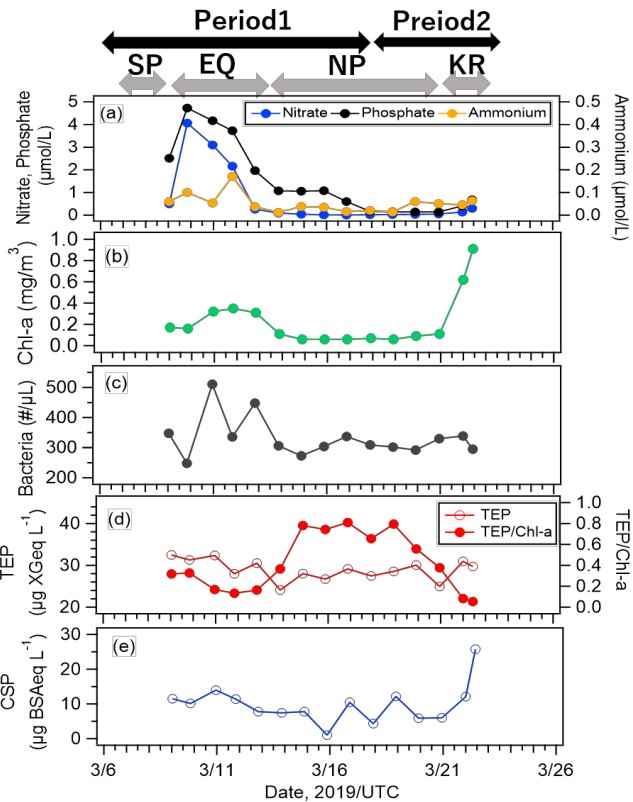

**Figure 6.** Time series of concentrations of (a) nutrients, (b) Chl-*a*, (c) bacteria, (d) TEPs and normalized TEP obtained by dividing TEP concentration by Chl-*a* concentration, and (e) CSPs at the South Pacific subtropical region (SP), equatorial upwelling region (EQ), North Pacific subtropical region (NP), and south of Kuroshio Extension (KR).





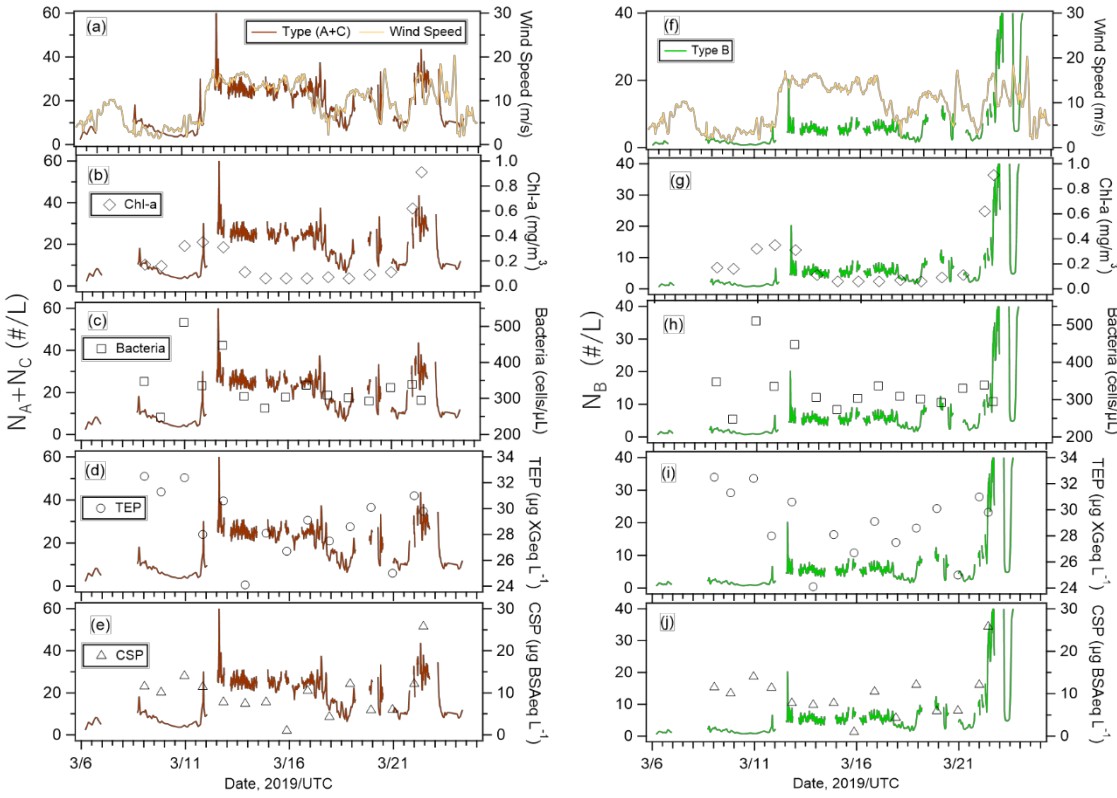

**Figure 7.** Time series of number concentrations of Types A+C particles and (a) wind speed, and concentrations of (b) Chl-*a*, (c) bacteria, (d) TEPs, and (e) CSPs. Time series of number concentrations of Type B particles and (f) wind speed, and concentrations of (g) Chl-*a*, (h) bacteria, (i) TEPs, and (j) CSPs.



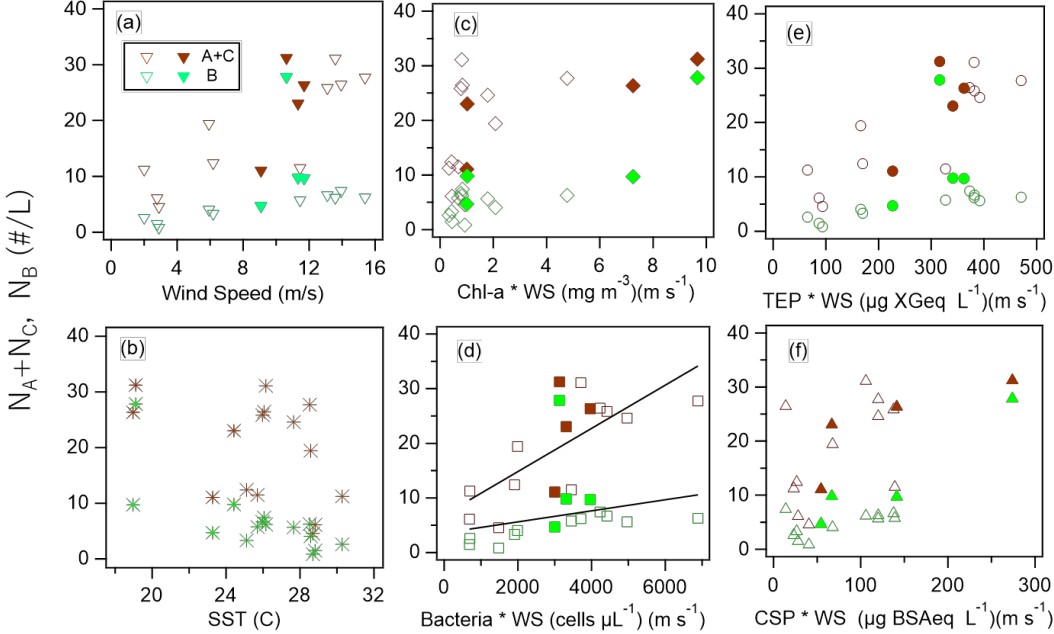

**Figure 8.** Number concentrations of Type A+C and Type B particles and (a) wind speed, (b) SST, (c) product of wind speed
(WS) and Chl-*a* concentration, (d) product of WS and bacteria concentration, (e) product of W and TEP concentration, and (f)
product of WS and CSP concentration. Open markers indicate Period 1 and solid markers indicate Period 2. The black lines in
Fig. 9d are regression lines for all Type A+C data and for all Type B data as an example.





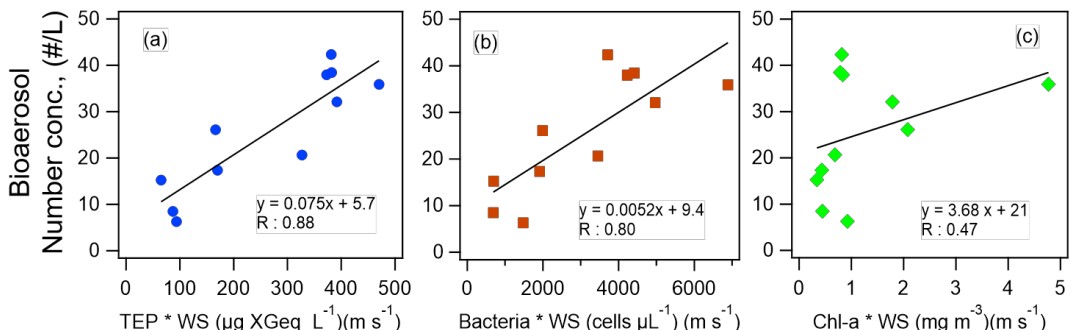

**Figure 9.** The scatter plots and equations of number concentrations of bioaerosols in the atmosphere as functions of the product
of wind speed (WS) and the bioindicators of (a) TEP concentration, (b) bacteria concentration, and (c) Chl-*a* concentration.