# Peer review of "Fluorescent biological aerosol particles over the central Pacific Ocean: covariation with ocean-surface biological activity indicators"

_Atmospheric Chemistry and Physics, 2021_

## Author Comment (AC1)

*Reviewer #1:*

*The manuscript by Kawana et al. studies the relationship between ambient bioaerosols (measured using single-particle autofluorescence) and various chemical and biological parameters of aerosols and sea water. The data was collected through a transect on the Pacific Ocean. The authors find various correlations between the concentration of fluorescent aerosol particles and biogenic sea water proxies and windspeed. From these observations, various parameterizations are developed to predict the concentration of FAPs in the marine atmosphere using biogenic sea water proxies and wind speed.*

*The manuscript is well written and the analysis appear (at least for main parts) sound and thorough. The study presents new and useful observational data which can be used to develop model parameterizations. My main criticism relates to the statistical significance of the presented findings and the developed parameterizations. Due to the limited number of samples (which is natural due to the comprehensive analysis needed) some of the equations and statements seem to come with a very large uncertainty. However, a thorough error analysis and discussing is currently missing and should be added before being finally published in ACP. This and further detailed comments are given below.*

We would like to thank the reviewer for providing valuable comments and better perspective on our work. Our responses are given below. All changes are shown in color in the revised manuscript.

*Introduction: Many of the references are selective. I would suggest to add "e.g." before them to indicate that the references are just a selection.*

We corrected as suggested in the revised manuscript.

*Page 1, line 26: could -> can*

We corrected as suggested in the revised manuscript.

*Page 3, line 7: More details on the aerosol inlet are needed. For example, was a size cut used? What was the length of the inlet? Was it heated or was the sample flow dried? What was the average RH at the inlet of the WIBS?*

We have added the following sentences about the inlet:

"For WIBS-4A, ambient particles were sampled using a total suspended particles (TSP) inlet (URG, URG-2000-30DG) installed at ~18 m above sea level on the compass deck. Particles passed through a stainless-steel tube with a volumetric flow rate of 30 L min$^{-1}$, to the research room and a large diameter Nafion tubing drier (Perma Pure Inc., MD-700) to dry the sample flow. Before passing through the dryer, the flow was iso-kinetically separated into two lines. One line was for a bypass vent flow, whose volumetric flow rate was 29 L min$^{-1}$, while the other passed through the drier was further separated into relative humidity (RH) monitor, WIBS-4A, and other instruments in the conductive silicone tube. During the cruise, the RH remained

below 40% at room temperature." (Page 3, Lines 7-13).

*Page 4, line 5: Since the instrument does not only count but also sizes the particles by optical means, I would suggest to replace OPC by OPSS: optical particle size spectrometer.*

We appreciate this comment. Though *OPSS* may indeed be an accurate description, we will keep *OPC* in the manuscript as the product name is "Handheld optical particle counter".

*Page 5, line 6: Why was a starting (or arrival) altitude of 500 m and not sea level (or inlet height) chosen for the trajectory analysis? I could imagine that it could make a difference for some of the trajectories.*

The starting altitude lower than 500 m increased the number of cases where the air parcel hit the sea surface and discontinuity occurred. Calculations were also carried out with a starting altitude of 1000 m and there was no essential difference in the results. Therefore, only the results with the starting altitude of 500 m were presented. As described in the manuscript, the timing of change of air masses suggested by the trajectories from 500 m was consistent with the timing of the increase in the ozone concentration, and thus the results of the trajectories are considered to be reasonable.

*Figure 1 and 2: To save space you could consider to move those figures to the supplement as well since they are mainly used to justify the two periods and don't provide much information concerning the biological aerosol particles.*

We partly agree with the comment, but after considering that Fig. 1 includes information on the cruise track and air mass origins and that Fig. 2 includes information on the meteorological condition (especially wind direction and wind speed are important) we decided to keep the figures in the main text.

*Figure 3 and 4: It would be helpful to also indicate the different periods here.*

We corrected as suggested in the revised manuscript.

*Figure 4b: There are two striking peaks for ABC particles (right at the beginning and close to the 11th of March. Is this linked to ship pollution?*

The first peak just after the beginning of measurement was excluded because of a possible influence of ship pollution or instability of the instrument. The second peak on 11-12 March was not removed as the OPC data were not regarded suspicious. The data points of CO and ozone were less; this was because the ozone level was low (~10 ppb) and the data filtering criterion to remove hourly data with the standard deviation of 1-min data exceeded 10% of the hourly average was easily met, while in reality the air mass was not affected by the ship exhaust. As such we could not identify the cause of the second peak and therefore included it in the plot.

*Page 6, last paragraph and Figure 5: The number of samples from the Bioplorer analysis is limited to only 10 samples, correct? Or the Bioplorer data been averaged to match the filter chemical analysis? An overview table with the performed analysis, instruments and their temporal resolution / number of samples would be useful (e.g. within the method part).*

Yes, ten samples were obtained during the cruise; manual workload required for each sampling and analysis (e.g., setting the gold-coated filter, flow control, retrieval, and post-analysis) was high and thus the number of samples was limited (information added in Table 1). Even under this limitation, however, the temporal/spatial variabilities of the target particles were well captured as demonstrated in Figure 5. For each sample, fluorescent particles were separately counted for 30 sub-gridded areas of the filter with the Bioplorer. Also, the total count followed Poisson distribution and provided information of random uncertainty. We added the following sentence and Table 1 in the revised manuscript:

"The observed parameters are summarized in Table 1." (Page 3, Line 4).

"The air samples for Bioplorer were collected with a flow rate of ~1 L min$^{-1}$ directly onto gold-coated membrane filters for 1-2 hr (pore size: 0.4 µm, KB-VKF02, Koyo Sangyo Co., Ltd., Tokyo, Japan) where epifluorescence detection was conducted." (Page 3, Lines 26-28).

"Although the samples for the Bioplorer were not measured repeatedly, the random uncertainty of the count was estimated to be 6%, assuming that the count followed the Poisson distribution." (Page 7, Lines 17-19).

Table 1. Summary of observations during the cruise (*batch sampling/offline analysis).

| Observed parameters | Instruments | Time resolution | Number of data/samples |
|---|---|---|---|
| Fluorescent particles | WIBS-4A | 1 hr | 456 |
| Ozone, CO | Ozone monitor CO monitor | 1 hr | 302 |
| Number-size distribution | OPC | 1hr | 234 |
| *Chemical composition | High-Volume air sampler | 24-72 hr | 9 |
| *Biological particles | Bioplorer | (1-2 hr, sampling duration) | 10 |
| *TEP, CSP, Bacteria, Chl-a, and Nutrients (surface seawater) | See text | See text | 15 |

*Page 7, line 15ff and Figure S2: The correlations are for certain parts driven by a few data points only. It should be stated that the statistics is still limited due to the low number of samples.*

Error bars were added for TEP and CSP, whose uncertainties were estimated from

reproducibility. Also, in the revised manuscript, it was clearly stated that the correlation was evaluated with a limited number of samples.

*"It should be noted that the analysis was made with a limited number of data points."* (Page 8, Line 18).

*One major weak point of this study is that no measurement uncertainties or other statistical parameters (like standard deviations or percentiles) are given. The study would significantly improve if uncertainty bars would be added to the key figures or given in tables and discussed within the text. In Sect. 3.4, the presented parameterizations (Eq. 1-3) need more profound statistical analysis. This is important because readers and potential users need to be aware of the potentially large uncertainties involved here. For example, it can be easily seen that the relationship between bioaerosol concentration and the chl-a\*ws parameter is driven essentially by one point (Fig. 9a). I would therefore like to ask the authors to add an uncertainty analysis (incl. error bars in the main figures, especially Fig. 9). The parameters of Equations 1-3 should include uncertainties. One approach could be to use a bivariate weighted fit (see e.g. York et al., 2004). Please also check the units of all coefficients carefully (e.g. the intercepts need units).*

We appreciate this valuable comment. We thoroughly estimated uncertainties of the quantities studied in this manuscript and revised sentences as follows:

"Uncertainties in the observation data and equations include random errors such as temporal variation and from measurements, and systematic errors due to the instruments. Among these, the random uncertainties (TEP, CSP, and WIBS data) are shown as error bars in the figures." (Page 11, Lines 1-3).

Although repeated measurements were not performed for Chl-*a*, the precision and values of coefficient of variation (CV) are mentioned in the revised manuscript (Page 7, Lines 26-29) and the range of measured values for bacteria are shown in Fig. 6.

In Fig. 9, the equations were revised to those from orthogonal regression lines, and the uncertainties of slopes and intercepts were included as follows.

$y$ (particles $L^{-1}$) = (0.076±0.014) ·[TEP, µg XGeq $L^{-1}$] · WS (m $s^{-1}$) + (5.4±4.1) (R: 0.88)    (1)

$y$ (particles $L^{-1}$) = (0.0052±0.0013) ·[bacteria, cells $µL^{-1}$] · WS (m $s^{-1}$) + (9.3±4.8) (R: 0.80)   (2)

and

$y$ (particles $L^{-1}$) = (20.0±19.0) · [Chl-a, mg $m^{-3}$] · WS (m $s^{-1}$) + (0.29±25) (R: 0.47) (3)

(Page 10, Lines 24-27, and caption in Fig. 9).

"It should also be noted when using the equations that additional systematic uncertainty of the factor of ~2 would be present considering the difference between the number densities derived from WIBS-4A and Bioplorer (see Sect. 3.2). Future studies with a larger number of samples

are warranted for full validation." (Page 11, Lines 3-5)

*I would recommend that the authors include in the presentation and discussion of their results the study by Santander et al (2021), who recently characterized the performance of the WIBS in controlled sea spray experiments.*

As suggested, we added the following sentence.

"Our interpretation is consistent with Santander et al. (2021), where the dominance of bacteria in fluorescent sea spray aerosols was confirmed with laboratory-generated aerosol particles and seawater." (Page 11, Lines 17-18)

*Why hasn't the size information of the WIBS not been fully exploited? How did the particle size of the different fluorescent particle classes (e.g., Fig. 4) change with time? The authors could think to add an overview table to the manuscript or the SI that gives averages parameters (and their variation) of the different classes. Figure S1 does show that certain FAP classes have much larger diameters. Could these be more primary bioaerosols? See also paper by Santander et al. (2021).*

We added a figure summarizing the size differences of fluorescent particles between Periods 1 and 2 (Figure S1e). They are shown individually for the particle classes emitting fluorescence in a single band (types of A, B, and C) and multiple bands (types of AB, BC, and ABC). They did not show a significant difference between the two periods; the average particle sizes for types A, B, and C during Period 1 (Period 2) ranged from 1.35–1.47 µm (1.17–1.55 µm), and those for types AB, BC, and ABC during Period 1 (Period 2) ranged from 1.93–3.07 µm (1.69–3.01 µm), respectively.

We added the following sentence.

"The size was slightly smaller than that reported for the type A particles (identified as bacteria), ~2–3 µm, generated with coastal seawaters (Santander et al., 2021)." (Page 6, Lines 29-30)

*The last paragraph of Sect 3.4 could be shortened and combined with the conclusions.*

As suggested, the last paragraph of Section 3.4 of the previous manuscript was fully moved to Conclusions.

*Data availability: It would be beneficial to the reader and the community if the data behind this study can be found on a public repository to follow the FAIR principles of data sharing (see data policy of ACP, https://www.atmospheric-chemistry-andphysics. net/policies/data_policy.html).*

We are now submitting the data used in this paper to the PANGAEA data repository, and the data will be available soon. We added the following sentence.

"The data discussed in this manuscript are available through the following websites. Cruise

data: https://doi.org/10.17596/0001976, JAMSTEC (2019) MIRAI MR18-06 Leg4 Cruise Data and PANGAEA data repository – Data Publisher for Earth & Environmental Science: https://doi.pangaea.de/10.1594/PANGAEA.xxxxxx" (Page 12, Lines 23-26; link to be finalized before revision).

*In general, the time series figures could be increased in size (mainly width). Especially for Fig. 3, 4, 6 and 7 it is hard to discriminate all the features.*
As suggested, the size in the figures was increased.

*Figure 8: Similar to the comment above, what kind of regression was used? It should be an orthogonal one. Please add this information to the text or figure caption(s).*
As suggested, we switched to the orthogonal fittings and this was mentioned in Fig. 8.
"The black lines in Fig. 8d represent the orthogonal regression lines for all Type A+C data and for all Type B data as an example."

*Figure S2: As mentioned above, it is also clear from this figure that the correlation coefficients are driven by a few outliers. This should be properly discussed within the text.*
As suggested, we added the following sentence.
"It should be noted that the analysis was made with a limited number of data points. If the last data point during the cruise was omitted where both terrestrial and biogenic sources seemed to contribute, the correlations among all biological indicators (TEP, CSP, Chl-*a*, bacteria) were not remarkable (R: ~0.6). As it has been shown that there is a high correlation between TEP concentration and *Synechococcus* abundance in the oligotrophic ocean (Zamanillo et al., 2019), the contribution to the formation of marine gel particles will vary among phytoplankton communities. Our cruise observation was conducted over several ocean regions, and nutrient concentrations (Fig. 6a) and phytoplankton community composition (Fig. S5) varied widely. Such variations may have reduced the relationships between the bioindicators in the entire observation area. " (Page 8, Lines 18-24)

*Figure S5: This is an interesting figure (which could be moved to the main manuscript?) since it shows that the contribution of phytoplankton species clearly changed through the cruise. Is there any link between the WIBS particles classes with the phytoplankton species? Or is this not to be expected? The authors could test if they see a relationship of intensive parameters from the WIBS (e.g. ratios of particle classes) to the phytoplankton species contribution.*
We thank the reviewer for this comment.
Although tried, we did not find a clear relationship between Chl-*a* concentration calculated by the contribution from each phytoplankton species and each class of fluorescent bioaerosols in the WIBS. Capturing the relationship is difficult with the limited number of samples in this study. For bioaerosol production, not only the community of phytoplankton species but also TEP

production associated with different nutrient environments and wind-driven effects are also important. More future studies are required.

*Other changes:*

1) We excluded all observation data from 12:23 on 07 March 2019 12:23 to 00:10 on 09 March 2019, due to the Exclusive Economic Zone (EEZ).
2) We changed the colors in the markers in Fig.7, Fig.8, Fig.S3, and Fig.S4 for clarity.

*Reference:*

Santander, M. V., B. A. Mitts, M. A. Pendergraft, J. Dinasquet, C. Lee, A. N. Moore, L. B. Cancelada, K. A. Kimble, F. Malfatti, K. A. Prather.: Tandem Fluorescence Measurements of Organic Matter and Bacteria Released in Sea Spray Aerosols, *Envrion. Sci. Technol.,* 55, 8, 5171-5179, 2021.

---

## Author Comment (AC2)

*Reviewer #2:*

*In this manuscript, K. Kawana et al. investigate fluorescent biological aerosol particles over the central Pacific Ocean and propose equations to derive atmospheric bioaerosol number density in the marine atmosphere from a combination of biogenic proxy quantities and wind speed. These equations could help in the parameterization of models, and as such, this manuscript will make a nice addition to the literature. My only relatively major concern is the absence of discussion on analytical or sampling uncertainties, especially for the following parameters: nutrients, Chl-a, bacteria, TEP, and CSP. The rest of my comments (see below) are mostly minor.*

We would like to thank the reviewer for providing valuable comments and better perspective on our work. Our responses are given below. All changes are shown in color in the revised manuscript.

*Page 4, lines 7-9: "To avoid contamination from ship exhaust, the data points from the online measurements were screened using the same criteria that were applied to the operation of the pump of the high-volume air sampler". Have you checked whether this criteria of ±75° from the bow is stringent enough? A simple test is to check whether you have instances with sudden ozone titration (due to NOx emissions from the ship exhaust).*

We used the OPC data to evaluate this criterion. When the same criteria were applied for the OPC data, the sudden increase/decrease events were removed so we concluded the criteria of ±75° from the bow was acceptable basically. The screening criteria for ozone data were developed independently as the location of the inlet was different from that for the aerosol measurements. Thus, the direct comparison of the criteria was not considered straightforward.

[Figure]

*Page 4, lines 10-13: surface seawater sampling was carried out with a bucket but you only analyzed 200 mL of water (if I understood correctly). Is this volume of water (200 mL)*

*representative of what was in the bucket (homogeneous sample)? Did you collect replicate filters? If so, please indicate the results as error bars in the figures. If no replicate samples were collected, could you please at least discuss analytical uncertainties and report them as error bars in the figures? I'm simply wondering if the temporal variation you describe later in Figures 6-7 is significant or if it's just noise.*

We added the following sentence describing uncertainties and precision in the revised manuscript:

"For the analysis of TEPs and CSPs, 200 mL of seawater out of ~10 L collected in a bucket was filtered onto a Whatman 0.4 µm Nuclepore hydrophilic polycarbonate membrane filter (Cytiva, Tokyo, Japan) where the particles were retained. By repeating this procedure, sample filters were made in triplicate." (Page 4, Lines 16-18).

We also described the representativeness of the data or related information on the nutrients, Chl-a, as well as TEP and CSP as follows:

"The analytical precisions of the nutrients and Chl-a concentrations were all <1%. For bacteria, ranges are shown for samples where duplicate measurements were made (Fig. 6c). For TEP and CSP, error bars represent one standard deviation from the repetitive analysis (Figs. 6d and 6e). They were all small enough to regard that their natural variations were captured by the observations." (Page 7, Lines 26-29).

*Page 5, lines 5-6: There should be a section on back-trajectories in the Methods. In addition: 1) which meteorological data did you use to generate the trajectories? 2) why did you use a starting altitude of 500 meters? 3) please also include the fact that you only generated trajectories twice a day (0600 and 1800 UTC according to Fig. 1 caption).*

Although considered, we decided to keep this description on the trajectory methods with the results together in section 3.1 for better readability.

1) As suggested, we added the following sentence.

"The used meteorological field was the Global Data Assimilation System with 1°×1° resolution (GDAS1) by the National Center for Environmental Prediction (NCEP) analyses." (Page 5, Lines 12-13).

2) The starting altitude lower than 500m increased the number of cases where the air parcel hit the sea surface and discontinuity occurred. Calculations were also carried out with a starting altitude of 1000m and there was no essential difference in the results. Therefore, only the results with the starting altitude of 500m were presented. As described in the manuscript, the timing of change of air masses suggested by the trajectories from 500 m was consistent with the timing of the increase in the ozone concentration, and thus the results of the trajectories are considered to be reasonable.

*3)* As already indicated in the Figure caption, we do not repeat this in the main text.

*Page 5, lines 10-13: Are these results in line with literature? See Bourgeois (2020) for example.*

We added the following sentence in the revised manuscript.

"Bourgeois et al. (2020) studied ozone concentrations over the similar latitudinal range and season and reported that the concentration increased in the north of ~10°N, further north than ~5°N for this study. It is likely that a strong northeasterly wind efficiently carried air masses with relatively high concentrations of ozone down to 5°N during our study period." (Page 5, Lines 23-26).

*Page 6, lines 1-5: How about the increase in type B particles at the end of the campaign?*

We added the following sentence.

"Type B particles increased at the end of the observation (Period 2), and the response with the wind was different from the behavior of types A and C. Effects from continental air masses during Period 2 (Fig. 1) or changes in the influential marine biota as aerosol sources were suspected. Detailed discussion is given in section 3.4." (Page 6, Lines 17-20).

*Page 6, lines 18-19: It's hard to tell with the log scale. Can you please add in the text the mean ± standard deviation before and after March 12 for each size range?*

We added the following sentence:

"The number concentrations of particles smaller (and larger) than 1 µm increased from 874±552 (and 913±680) $cm^{-3}$ during 10–12 March to 6903±650 (and 7436±2180) $cm^{-3}$ during 12–14 March." (Page 7, Lines 3-5).

*Page 6, lines 30-31: Please make it clear in the caption that the y-axis is different for the two instruments; it took me a while to realize that.*

We corrected as suggested in the revised manuscript.

*Page 7, line 8: "high again in the south of the KR on March 22". This is not obvious…I'd appreciate error bars on this Figure. Same comment for the rest of this section.*

We are unable to add error bars for nutrients as repeated measurements were not made. However, the Y axis in Fig. 6a is displayed as logarithmic for clarity and the values in the North Pacific subtropical region and the south of the Kuroshio Extension were described as follows.

"Nutrient concentrations were high in the EQ region, low in the NP region (especially nitrate was almost depleted throughout this region), and slightly increased again in the south of the KR region on 22 March. In detail, concentrations in the south of the KR region (nitrate: ~0.30, phosphate: ~0.07, ammonium: ~0.06 µmol $L^{-1}$) were higher than those in the NP region (nitrate: ~0.03, phosphate: ~0.05, ammonium: ~0.03 µmol $L^{-1}$). " (Page 7, Lines 30-33).

*Figure 1: Color on trajectories show air parcel altitude pressure (not altitude).*
We corrected as suggested in the revised manuscript.

*Figure 2: Very few data points for both ozone and CO before 3/12. Is that due to instrument issues or to pollution from the ship exhaust (wind out of the clean air sector)? If the latter, how did this pollution impact the representativeness of bioaerosol results presented in the manuscript? I'd appreciate a table showing the daily hours of operation of each instrument during the campaign to better appreciate the temporal representativeness of the samples.*
The scarce ozone and CO data before 12 Mar were due to a data screening criterion to remove hourly ozone data with standard deviations of 1-min data exceeding 10% of the average. Though NO titration was minor, the low-level ozone data (~10 ppb) were deleted because of natural variability with 1-min data in a range of 1 ppb (corresponding to 10%). We found this criterion too strict and will loosen it slightly in future studies. Thus, the issue was only for $O_3$ (and CO, where the same criterion was applied) and therefore bioaerosol data during this period were not removed similarly to O3. A new table (Table 1) will include information on temporal representativeness.

*Figure 4c: please make it clear in the caption that the y-axis is different for type C. I initially got confused.*
We corrected as suggested in the revised manuscript.

*Figure 6: please add the different regions on the map (Figure 1).*
We corrected as suggested in the revised manuscript.

*Figures 6-7: please add error bars!*
We corrected as suggested in the revised manuscript.

*Table S1: please clarify what "Zone" refers to. This should be added in the caption.*
We corrected as suggested in the revised manuscript.

*Other changes:*
1) We excluded all observation data from 12:23 on 07 March 2019 12:23 to 00:10 on 09 March 2019, due to the Exclusive Economic Zone (EEZ).
2) We changed the colors in the markers in Fig.7, Fig.8, Fig.S2, and Fig.S4 for clarity.

*Reference:*
Bourgeois, I., Peischl, J., Thompson, C. R., Aikin, K. C., Campos, T., Clark, H., Commane, R., Daube, B., Diskin, G. W., Elkins, J. W., Gao, R.-S., Gaudel, A., Hintsa, E. J., Johnson,

B. J., Kivi, R., McKain, K., Moore, F. L., Parrish, D. D., Querel, R., Ray, E., Sánchez, R., Sweeney, C., Tarasick, D. W., Thompson, A. M., Thouret, V., Witte, J. C., Wofsy, S. C., and Ryerson, T. B.: Global-scale distribution of ozone in the remote troposphere from the ATom and HIPPO airborne field missions, Atmospheric Chem. Phys., 20, 10611–10635, https://doi.org/10.5194/acp-20-10611-2020, 2020.